# Impact of Microencapsulation on *Ocimum gratissimum* L. Essential Oil: Antimicrobial, Antioxidant Activities, and Chemical Composition

**DOI:** 10.3390/foods13193122

**Published:** 2024-09-30

**Authors:** Angela Del Pilar Flores Granados, Marta Cristina Teixeira Duarte, Nathan Hargreaves Noguera, Dyana Carla Lima, Rodney Alexandre Ferreira Rodrigues

**Affiliations:** 1Department of Food Science and Nutrition, School of Food Engineering (FEA), University of Campinas (UNICAMP), Campinas 13083-862, São Paulo, Brazil; 2Microbiology Division, Multidisciplinary Center of Chemical, Biological, and Agricultural Research (CPQBA), UNICAMP, Paulínia 13148-218, São Paulo, Brazil; mduarte@cpqba.unicamp.br; 3Natural Products Chemistry Division, Multidisciplinary Center of Chemical, Biological, and Agricultural Research (CPQBA), UNICAMP, Paulínia 13148-218, São Paulo, Brazil; nogueraengenharia@gmail.com (N.H.N.); dyanaengalimentos@gmail.com (D.C.L.)

**Keywords:** cashew gum, inulin, maltodextrin, natural antimicrobial, spray-drying

## Abstract

*Ocimum gratissimum* (OG) is a species rich in essential oils (EO), which is known for its antimicrobial and antioxidant properties. This study aimed to encapsulate the essential oil of *Ocimum gratissimum* (OGE), determine its chemical composition, and evaluate its antioxidant and antimicrobial activities against six pathogenic bacteria, comparing it with the free essential oil (OGF). The EO was extracted by hydrodistillation using a Clevenger-type apparatus, and an oil-in-water emulsion was prepared using a combination of biopolymers: maltodextrin (MA), cashew gum (CG), and inulin (IN). The chemical profile was identified using gas chromatography–mass spectrometry (GC–MS). Antioxidant activity was assessed using the Oxygen Radical Absorbance Capacity with fluorescein (ORAC-FL) method, while the Minimum Inhibitory Concentrations (MIC) and Minimum Bactericidal Concentrations (MBC) were determined by the microdilution method. Microparticles were formed using the spray-drying method, achieving an encapsulation efficiency of 45.2%. The analysis identified eugenol as the main compound both before and after microencapsulation. The OGE microparticles demonstrated high inhibitory and bactericidal effects against *S. aureus*, *S. choleraesuis*, and *E. coli*, with MIC values of 500 µg·mL^−1^ and MBC values of 1000 µg·mL^−1^, as well as antioxidant activity of 1914.0 µmol-TE·g^−1^. Therefore, it can be inferred that the EO of OG maintained its antimicrobial and antioxidant effects even after microencapsulation by spray-drying, making it a promising natural ingredient.

## 1. Introduction

Foodborne diseases continue to pose a significant challenge for the food industry. Despite the availability of control measures and increased knowledge about pathogenic bacteria, these diseases still result in mortality and financial losses [1,2,3]. The food industry commonly employs preservatives to reduce the presence of foodborne pathogens; however, their adverse effects, such as allergic reactions, raise concerns about human health. Consequently, the search for natural additives has gained the interest of many consumers [3,4]. Studies have shown that natural antimicrobial preservatives derived from various sources, such as plants, animals, microorganisms, algae, bacteria, and fungi, have proven effective in ensuring food safety [5,6]. Essential oils are volatile compounds extracted from medicinal and aromatic plants, which have demonstrated antibacterial and antioxidant properties due to their complex chemical composition. This activity is based on the actions of their major compounds or interactions among them, which are key in exerting their bioactive properties [2,7,8]. 

*Ocimum gratissimum* L. belongs to the Lamiaceae family and is commonly known as basil, wild basil, and clove basil [9,10]. Native to Asia and South Africa and subspontaneous in Brazil, its essential oil is characterized by a high eugenol content, a key chemical compound in its composition [10,11]. Several studies have demonstrated that essential oil possesses antioxidant [10,12,13], anti-inflammatory [14], antifungal [15], and antibacterial properties [11,16,17,18]. However, essential oils are unstable, exhibit low water solubility, and are easily degraded in the presence of oxygen, light, moisture, and heat. These factors compromise their biological activity and hinder their preservation during storage [19,20]. Microencapsulation is a process in which small particles or droplets are encapsulated within a matrix or coating, offering a viable solution to protect essential oils from degradation, increase their stability, provide controlled release, and optimize their antimicrobial and antioxidant actions [3,4,21]. In recent years, research on the microencapsulation of plant-derived essential oils has grown significantly, driven by the interest in leveraging their benefits and expanding their industrial applications [3]. This technology has proven to be a promising approach to maximizing the advantages of essential oils and overcoming the limitations of their free form, such as volatility and sensitivity to environmental factors [22,23,24]. Despite recent advances, few studies have specifically addressed the encapsulation of *Ocimum gratissimum* essential oil and its effects on antimicrobial and antioxidant activities. Since this oil has demonstrated biological properties in its free from, its microencapsulation could expand its use in food and pharmaceutical products. Although some studies have explored this essential oil in processes such as nanoemulsions [1] and nanoparticles [25,26], further research is required to fully investigate the potential benefits of its microencapsulation.

The spray-drying method is widely used due to its efficiency, speed, low cost, and ease of application at the industrial level. Additionally, it allows for a flexible selection of biopolymers, which play a crucial role in the stability of microparticles and encapsulation efficiency [21,27]. Maltodextrin is one of the most used materials because of its relatively low cost, low viscosity, good solubility, and neutral taste and aroma. However, due to its low emulsifying and oil retention capacity, it is necessary to combine it with other polymers to achieve higher efficiency in microencapsulation and volatile retention [21,28]. Cashew gum, an exudate obtained from the *Anacardium occidentale* L. tree, exhibits high solubility and good emulsifying properties [29]. Native to Northeastern Brazil and introduced to Africa and India [30], this biopolymer has physicochemical characteristics similar to those of gum Arabic, and due to variations in price, availability, and quality, it has been suggested as an alternative for its partial or total replacement [1,31]. Inulin is a biopolymer composed of a linear chain of fructose monomers with a terminal glucose unit, offering both nutritional and technological properties. Interest in this material has grown due to its numerous benefits, such as its prebiotic nature, resistance to pH variations, stabilizing capacity, and potential as a slow-release drug delivery medium [32,33].

The objective of this research was to analyze the effect of microencapsulation on the chemical composition of *Ocimum gratissimum* L essential oil and to compare its antimicrobial and antioxidant activities before and after microencapsulation using six pathogenic bacterial strains.

## 2. Materials and Methods

### 2.1. Materials

The *Ocimum gratissimum* L. plant was cultivated at the Experimental Field of CPQBA-UNICAMP and is part of the Medicinal and Aromatic Plants Collection (CPMA) under voucher number UEC 121.407–CPMA 446. The biopolymers used included purified cashew gum from *Anacardium occidentale* L., obtained from raw exudate donated by EMBRAPA (Pacajus-Ceará, Brazil), maltodextrin MorRex^®^ 1910 with dextrose equivalent DE 10, donated by Corn Products (Mogi Guaçú, Brazil), and native inulin Frutafit^®^ IQ (DP ≥10), donated by Sensus (São Paulo, Brazil). The use of the *A. occidentale* species, a native Brazilian plant, was authorized following a request for Authorization of Access to Genetic Heritage in the National System for the Management of Genetic Heritage and Associated Traditional Knowledge (SisGen), under n^o^. AF20FCA. The emulsifiers Tween 80^®^ and Tween 20^®^ (Oxiteno, Synth, São Paulo, Brazil), as well as other chemical reagents used, were of analytical grades (P.A).

### 2.2. Methods 

#### 2.2.1. Purified Cashew Gum 

The batch of cashew gum was purified according to the methodology described by Rodrigues and Grosso [34], with modifications. The raw exudate was ground in an industrial blender (Laboratory Blender, Brazil) and subsequently dissolved in distilled water at a 1:2 (*w/v*) ratio (exudate:water). The solution was then subjected to gravity filtration to remove impurities such as trunk residues and sand. Subsequently, this solution was mixed with 96% ethanol (96° Gay-Lussac) at a 1:8 (*v/v*) ratio (solution:ethanol) to promote gum precipitation, which was immediately filtered again using gravity filtration. The purified gum was dried in an oven with forced air circulation at 45 ± 2 °C for 24 h. It was then ground in a multi-purpose mill (Model A11, Janke and Kunkel, IKA Labortechnik, Germany) to obtain the gum in powder form. 

#### 2.2.2. Essential Oil Extraction 

The essential oil was extracted using the method described by Duarte et al. [35], with some modifications. The extraction was carried out from fresh leaves by hydrodistillation at a 1:3 (*w/v*) ratio of plant material to distilled water, using a Clevenger-type apparatus for 3 h. The essential oil was then collected and treated with anhydrous sodium sulfate to remove excess water, after which it was stored at 5 °C in an amber flask. 

#### 2.2.3. Emulsion Preparation and Microencapsulation by Spray-Drying 

MA, CG, and IN were weighed in equal parts (1:1:1). These biopolymers were then dissolved in distilled water at 25 °C at a concentration of 30% (*w/v*) biopolymers: water. The biopolymers were prepared one day before use to ensure complete dissolution and kept refrigerated at 8 °C. Subsequently, the essential oil was added to the biopolymer solution at a concentration of 20% (*w*/*w*), along with the surfactants Tween 80 and Tween 20 in equal parts (1:1), at a concentration of 1% (*v*/*w*) relative to the total solids. The ingredients were homogenized at 24,000 rpm for 3 min using an Ultra-Turrax (Model T25, IKA, Staufen, Germany). The final emulsion was kept under mechanical stirring and spray-dried using a Mini Spray Dryer (B-290, Buchi^®^, Flawil, Switzerland) under the following conditions: inlet air temperature: 150 ± 2 °C; outlet air temperature: 90 ± 2 °C; airflow rate 35 m^3^·h^−1^; peristaltic pump rate 4.5 mL·min^−1^; and atomizing air 600 L·h^−1^. The dried microparticles were collected and stored in an amber container under refrigeration until further analysis.

#### 2.2.4. Moisture Content, Water Activity (aw), Encapsulation Efficiency (EE), Particle Size, and Scanning Electron Microscopy (SEM) 

The moisture content of the microparticles was determined by direct drying in an oven at 105 °C until constant weight was achieved, following AOAC guidelines [36]. Water activity was measured in triplicate using a water activity meter (Aqualab 3TE, Meter Group, Pullman, WA, USA) at 25 ± 0.5 °C. 

Encapsulation efficiency was calculated using Equation (1), where surface oil and total oil were determined according to the method of Bhandari, D’arcy, and Padukka [37], with modifications. A mixture of 3 g of microparticles and 20 mL of hexane was manually agitated for 10 min. The solution was then filtered through a Whatman No. 1 filter and rinsed with 10 mL of hexane three times. The solvent was evaporated using a rotary evaporator (Rotavapor R-100, Buchi, Flawil, Switzerland) at 45 °C, and the sample was placed in a vacuum oven for 5 min before being weighed on an analytical balance. Total oil determination was based on the total volatile oil retained both externally and internally in the microparticles. A sample of 10 g of microencapsulated essential oil was dissolved in 150 mL of distilled water. The essential oil was then recovered, and its volume was measured and multiplied by the density of 0.9791 to calculate the mass of the recovered oil.
(1)$$\%EE=100\times \left(\frac{total\;oil-surface\;oil}{initial\;oil}\right)$$

Particle size was measured using a light scattering instrument (LV 950-V2, Horiba, Kyoto, Japan) following the methodology of Fadini et al. [38]. The samples (0.5 g per 100 g^−1^) were suspended in absolute ethanol, and the mean diameter was expressed as D_4:3_. Polydispersity was analyzed by the span value, calculated according to Equation (2), where D_0:1_; D_0:5_ and D_0:9_ correspond to the diameters at 10, 50 and 90%, respectively, of the cumulative particle size distribution.
(2)$$span=\left(\frac{D_{0.9}-D_{0.1}}{D_{0.5}}\right)$$

The morphology of the microparticles was observed using Scanning Electron Microscopy with a LEO 440i Scanning Electron Microscope (Electron Microscopy, Oxford, UK). The samples were mounted on stubs and coated with gold using a K450 sputter coater (EMITECH, Kent, UK) for 3 min at a rate of 0.51 Å and an acceleration voltage of 10 kV. The observations were made at magnifications of 2500× and 10,000×. Image acquisition was performed using LEO software, version 3.01.

#### 2.2.5. Chemical Characterization by GC–MS Analysis 

The identification and quantification of chemical compounds in OGF and OGE were analyzed using an HP-6890 gas chromatograph equipped with an HP-5975 mass selective detector and an HP-5MS capillary column (30 m × 0.25 mm diameter × 0.25 µm film thickness), all from Agilent Technologies (California, USA). The electron ionization system (70 eV) was equipped with a split/splitless injector, operated in split mode with a ratio of 1:40. The temperature conditions were as follows: injector = 220 °C; column = 60 °C; heating ramp rate = 3 °C·min^−1^; and final temperature of 240 °C for 7 min, with the mass detector (MS) at 250 °C. Helium was used as the carrier gas at a flow rate of 1 mL·min^−1^. The essential oil was dissolved in ethyl acetate (20.0 mg·mL^−1^), and the retention index (RI) was determined by co-injection of a mixture of aliphatic hydrocarbons (C_8_−C_22_) using the Van den Dool and Kratz formula. The constituents of the essential oil were identified by comparing their RI and mass spectra with the literature data from Adams [39] and the NIST 11 library. 

#### 2.2.6. Antioxidant Activity

The antioxidant activity was determined using the ORAC-FL method with Fluorescein, following the adapted methodology described by Salvador et al. [40]. The experiments were conducted in 96-well microdilution plates after preparing stock solutions of the samples (50 mg·mL^−1^) in phosphate buffer/DMSO (99:1, *v/v*). Trolox was used as the reference standard at concentrations of 12.5, 25, 50, 100, and 200 µM. The readings were taken using a fluorescence filter (485–528 nm), monitoring every 2 min for 70 min at a temperature of 37 °C. The samples considered included OGE, OGF, and control microparticles without essential oil (BS). The results were expressed as µmol of Trolox equivalent (TE) per gram of essential oil (µM TE·g^−1^).

#### 2.2.7. Antibacterial Activity 

The antibacterial activity of OGF and OGE was studied against the bacteria *Salmonella choleraesuis* (ATCC 10708), *Escherichia coli* (ATCC 11775), *Staphylococcus aureus* (ATCC 6538), *Listeria innocua* (ATCC 33090), *Pseudomonas aeruginosa* (ATCC 13388) and, *Bacillus cereus* (CCT 2576), obtained from the Brazilian Collection of Environmental and Industrial Microorganisms–CBMAI, at CPQBA/UNICAMP. The experiments were conducted according to the methodology described in CLSI [41]. The bacteria were suspended in sterile saline solution (0.85% NaCl; *w/v*) to obtain an absorbance between 0.08 and 0.10 at 625 nm, which corresponded to the 0.5 McFarland turbidity standard (1.5 × 10^8^ CFU·mL^−1^).

#### 2.2.8. Minimum Inhibitory Concentration Determination 

The determination of the Minimum Inhibitory Concentration was used to assess the antibacterial activity of OGF and OGE. The samples were solubilized in Mueller–Hinton broth (MHB) containing 2.5% dimethyl sulfoxide (DMSO) and a 0.1% Tween 80 solution in water. Solutions were prepared with a maximum concentration of 2000 µg·mL^−1^, considering the mass of essential oil present in the total microparticles for the OGE sample. The 96-well microplates were prepared with 100 µL of MHB, 100 µL of bacterial inoculum diluted in the medium to 10^4^ CFU·mL^−1^, and 100 µL of each sample (OGF and OGE) at different concentrations in each well (2000–15.6 µg·mL^−1^). The contents of the wells were homogenized. Controls included chloramphenicol antibiotic, control microparticles without essential oil, culture medium (MHB), and bacterial growth controls. The microplates were incubated at 37 °C for 24 h, and the MIC was detected by adding 50 µL of TTC (2, 3, 5-triphenyl tetrazolium chloride, (Merck, São Paulo, Brazil) at 0.1% to the wells, followed by re-incubation at 37 °C for 3 h. All experiments were performed in triplicate. MIC values were defined as the lowest sample concentration capable of preventing visible bacterial growth. For the determination of Minimum Bactericidal Concentration, 10 µL of sample from each well corresponding to the MIC and higher concentrations was transferred to nutrient agar plates, surface plated, and incubated at 37 °C for 24 h. MBC was defined as the lowest concentration of EO capable of inhibiting bacterial growth by 99.9%, being considered bactericidal. If growth was observed, the activity was considered bacteriostatic.

#### 2.2.9. Statistical Analysis 

All assays were performed in triplicate, and the results were expressed as the mean ± standard deviation. Data were analyzed using one-way analysis of variance (ANOVA). Tukey’s test, with a 5% significance level, was used to compare significant differences between the mean values using Minitab (v.17 Statistical Software, Minitab, Inc., State College, PA, USA). 

## 3. Results and Discussion

### 3.1. Characterization of Microencapsulated Essential Oil 

The results related to the characterization of OGE microparticles using the three biopolymers, MA, CG, and IN, are presented in Table 1.

#### 3.1.1. Moisture Content, Water Activity, and Encapsulation Efficiency 

In this study, the moisture content of the dried microparticles was 3.91% ± 0.11. Despite the high temperature used in the spray-dryer (150 °C), a residual moisture content of around 4% remained in the particles, which is consistent with this process due to its rapid nature. This moisture level was similar to those reported by Pilicheva, Uzunova, and Katsorov [42] in lavender essential oil microparticles with gum arabic and maltodextrin obtained using the same drying method. The authors noted the effect of moisture levels in relation to the type and the concentration of biopolymers used, particularly emphasizing the importance of using MA with DE 10, which was also used in our study, contributing to the reduction in moisture levels in the microparticles. On the other hand, the food industry considers moisture levels between 3% and 4% acceptable within the specifications for most dried powders [43].

The aw of the microparticles was 0.25 ± 0.02, an important characteristic for extending shelf life during storage. The reduced water activity (aw < 0.40) indicates greater microbiological, physical, and chemical stability, resulting in higher quality products, which is highly applicable in the food industry [44].

The encapsulation efficiency using the combination of MA, CG, and IN in equal proportions was 45.2% ± 0.4. This value indicates the amount of essential oil that was encapsulated or retained within the microparticles formed by the combination of these biopolymers. Expressed as a percentage, this parameter reflects how the drying process conditions influenced the losses due to volatilization [45]. Similar studies have been reported by Fernandes et al. [33], who used combinations of CG and IN in different proportions for the microencapsulation of ginger essential oil, showing EE values ranging from 15.81% to 31.19%. The combination with a 1:1 ratio of CG to IN achieved the highest essential oil retention. Several parameters may have influenced the difference in results; in our study, the higher retention of essential oil may be related to the presence of maltodextrin in the formulation. Azhar et al. [46] emphasize that the selection of polymers is crucial for encapsulation efficiency in the spray-drying process, recommending the combination of biopolymers when a single polymer does not meet essential characteristics, such as solubility, emulsifying capacity, and adequate viscosity. Similarly, Pilicheva, Uzunova, and Katsorov [42] observed that lavender essential oil microparticles, utilizing gum arabic and maltodextrin, exhibited EE values ranging from 26.53% to 91.08%, with the EE of the 1:1 mixture of these two polymers being 44.10%. This result is close to the EE values obtained in our study, although other parameters may also have influenced volatile retention. According to Ré [47], volatile retention is influenced by the air inlet temperature during the drying process; as the temperature increases, retention also increases, while the amount of essential oil on the surface of the microparticles decreases.

#### 3.1.2. Particle Size and Scanning Electron Microscopy

The particle size and distribution, influenced by the type of polymers used, are critical factors as they affect both the appearance and flowability of the dry powder [46,48]. The particle size results showed a span value of 1.25 ± 0.05 and a D_4.3_ of 7.32 ± 0.81 µm. The D_10_, D_50_, and D_90_ values were 3.27 ± 0.04 µm, 6.29 ± 0.17 µm, and 11.11 ± 0.48 µm, respectively. The mean diameter was calculated and expressed as D_4.3_ (De Brouckere mean diameter), indicating the diameter of a sphere with the same volume as the particle. It should be noted, however, that particle size is not directly related to volatile retention. Nevertheless, for optimal rehydration of the dry powder, it is preferable to produce larger particles [48]. The microparticles exhibited high polydispersity, as indicated by the span value. Particles produced by the spray-drying method typically form a very fine powder, with the size dependent on the polymers used and the drying process conditions [21].

The external morphology of the microparticles was analyzed using scanning electron microscopy at magnifications of 2500× and 10,000×, as shown in Figure 1. At 2500× magnification (A-1 and A-2), with scale bars of 10 µm, the microparticles exhibit irregular shapes and surface agglomeration, providing better insight into their size distribution and variation. In contrast, images B-1 and B-2, taken at 10,000× magnification with scale bars of 1 µm, reveal that the microparticles possess a spherical shape and a rough surface, with no apparent cracks or fissures, indicating effective protection of the essential oil. These results are consistent with those of Fernandes et al. [43], who analyzed ginger oil microparticles with whey protein isolate and its combinations with MA and IN, observing a spherical and rounded morphology with indentations on the external surface. Beirão Da Costa et al. [32] highlight that the structure of microparticles can influence characteristics such as bulk density, volatile loss, and rehydration. These authors studied the microencapsulation of oregano essential oil with inulin and observed that the drying temperature (120, 155, and 190 °C) affected the formation of smooth or rough surfaces. As the temperature decreased, cavities appeared on the surface of the microparticles. This phenomenon occurs because, at high temperatures, rapid evaporation causes microparticle expansion, while at lower temperatures, the slow diffusion of water results in the shrinkage and collapse of the microparticles.

### 3.2. Chemical Characterization of Ocimum gratissimum Essential Oil 

In this study, the chemical profile of *Ocimum gratissimum* essential oil was analyzed using gas chromatography coupled with mass spectrometry. The chromatogram, shown in Figure 2, reveals five prominent peaks at retention times of 8.28 min (Peak 1), 21.25 min (Peak 2), 23.48 min (Peak 3), 25.97 min (Peak 4), and 27.15 min (Peak 5). These peaks correspond to the major compounds identified: cis-β-Ocimene (7.8%); Eugenol (66.4%); β-caryophyllene (3.1%); germacrene-D (5.6%); and farnesene (13.4%), which together account for approximately 96% of the total oil content. The complete chemical composition, including the 16 compounds identified, is detailed in Table 2. Similarities were found in studies by Melo et al. [49], who identified 19 compounds in the essential oil of *Ocimum gratissimum*, with eugenol being the major compound (74.8%), followed by 1,8-cineole (15.2%). Chimnoi et al. [16] identified 37 compounds, with six main ones: eugenol (55.6%); cis-ocimene (13.9%); γ-muurolene (11.6%); α-farnesene (5.6%); α-trans-bergamotene (4.1%); and β-caryophyllene (2.7%). Sartoratto et al. [50] characterized six compounds, with eugenol (93.9%) and germacrene-D (4.2%) being the most prevalent. These variations in the chemical composition of the plant indicate changes in secondary metabolism, influenced by factors such as cultivation methods, climate, harvest season, altitude, pathogen attacks, and harvest area [1,16,49]. Eugenol, a hydroxyphenylpropene, is the main chemical compound in the essential oil of *Ocimum gratissimum*. The quantity of eugenol can vary according to the season; eugenol was initially extracted from clove. This compound is widely recognized for its potent antioxidant and antibacterial properties [1,51]. Additionally, eugenol is considered safe by the World Health Organization (WHO) under the GRAS classification, allowing its use in various applications in the pharmaceutical, cosmetic, and food industries [51].

### 3.3. Chemical Characterization of Encapsulated Ocimum gratissimum Essential Oil

The microencapsulation of volatile compounds via spray-drying presents the challenge of evaporating water while stabilizing volatile substances such as essential oils [48]. In this study, 16 chemical compounds were identified in the essential oil of *Ocimum gratissimum*. However, after encapsulation, only nine of these compounds were detected in the OGE sample, as shown in Table 2. It was observed that in the OGE sample, there was a reduction in cis-β-ocimene (5.8%), germacrene-D (5.2%), and farnesene (11.4%), and an increase in eugenol (71.5%) and β-caryophyllene (3.4%), compared to the OGF. Compounds with percentages below 0.3% in the OGF sample were volatilized during the encapsulation process. Nevertheless, the results indicate that the five main constituents of the essential oil, both in the free OGF and encapsulated OGE forms, showed minimal differences in relative area percentage (%A) after the drying process, as illustrated in Figure 3.

These results are consistent with the findings of Nhan et al. [52], who observed changes in the chemical composition of lemongrass oil, with a reduction in myrcene and an increase in citral after microencapsulation. The authors concluded that the reduction in myrcene could be attributed to its high volatility, as low molecular weight monoterpenes tend to evaporate at elevated temperatures, reducing their concentration. In our study, the compounds α-thujene, sabinene, and myrcene, three monoterpenes that had concentrations of 0.1% in OGF, were volatilized in OGE. Radünz et al. [53] corroborate these findings by reporting a reduction in non-oxygenated compounds such as para-cymene and γ-terpinene in encapsulated thyme essential oil, while oxygenated compounds like linalool and camphor were better retained due to the formation of hydrogen bonds.

Reineccius [48] identifies three critical periods for the retention of volatile compounds during the drying process: the formation of droplets before the boiling point of water; the rapid loss of water from the wet particles; and the morphological development of particles after the water reaches its boiling point. In our study, the loss of volatiles may be related to the boiling point, as the compounds ethylbenzene, para-xylene, and α-thujene have boiling points of 136 °C, 138 °C, and 151 °C, respectively, while the temperature used in the spray-dryer was 150 °C. Studies by Van et al. [54] also observed changes in the proportions of essential oil constituents of *Citrus latifolia* after microencapsulation with maltodextrin. The authors concluded that these variations could be caused by the high-temperature spray-drying process, which evaporates low-boiling-point monoterpenes, while compounds with higher boiling points do not volatilize.

### 3.4. Antioxidant Activity 

The ORAC assay, based on hydrogen atom transfer, is an essential method for evaluating antioxidant capacity, especially in pure compounds and complex mixtures such as essential oils [55,56]. Figure 4 presents the ORAC results for the essential oil OGF, the microencapsulated OGE, and the control microparticles without essential oil, expressed in μmol of Trolox equivalents (TE) per gram of essential oil. According to Salvador et al. [56], ORAC values equal to or greater than 1000.0 µmol TE/g indicate strong antioxidant activity. The biopolymers used in the BS sample did not exhibit significant antioxidant activity. OGE showed significantly higher ORAC values (1914.0 µmol-TE/g) compared to OGF (1310.1 µmol TE/g), possibly due to the increased concentration of eugenol detected in the GC–MS analysis, contributing to the higher antioxidant capacity of OGE. Studies by Nehme et al. [57] indicate that the antioxidant activity of essential oils is directly influenced by their chemical composition, particularly phenolic compounds. In this context, Bentayeb et al. [55] demonstrated that the ORAC method is effective for measuring the antioxidant capacity of essential oils, which is strongly influenced by the contributions of individual oil components. The results for *Ocimum gratissimum* support previous studies that identified eugenol as the main compound in this plant, effective in neutralizing free radicals and preventing lipid oxidation [51]. Moreover, this essential oil has superior antioxidant properties compared to pure eugenol [58], making it crucial to protect the chemical compounds present in the essential oil.

In this study, the OGE sample preserved the antioxidant activity of *O. gratissimum* essential oil using a combination of biopolymers CG, MA, and IN in equal proportions without being affected by the conditions during the drying process. Maltodextrin acted as a matrix-forming agent [48]; cashew gum served as an emulsifier for stable emulsions [29], and inulin provided prebiotic benefits and additional technological properties [32]. These results are consistent with the findings by Lacerda et al. [59], where microcapsules encapsulated with starch, inulin, and maltodextrin combined with jussara pulp showed higher anthocyanin content and greater antioxidant activity using the ORAC method, especially in the mixtures of maltodextrin and inulin. The authors suggest that this effect may be related to the relative polarity of these carbohydrates. Similarly, Da Costa et al. [60] demonstrated that the encapsulation of oregano essential oil with different combinations of inulin, gelatin/sucrose, and rice starch preserved antioxidant activity for up to six months. Microparticles containing higher concentrations of inulin showed higher ORAC values, and the solid content in the initial emulsion was crucial for maintaining antioxidant activity.

### 3.5. Antibacterial Activity 

The antibacterial activity results of OGF and the OGE microparticles against pathogenic bacteria are presented in Figure 5 and Table 3. According to the classification by Duarte et al. [35], minimum inhibitory concentration values of up to 500 µg·mL^−1^ indicate strong inhibition; values between 600 and 1500 µg·mL^−1^ indicate moderate inhibition, and values above 1600 µg·mL^−1^ indicate weak inhibition. In this study, control samples, such as microparticles without essential oil, were used to assess whether the biopolymers employed had any inhibitory effect on the bacteria in addition to the antibiotic chloramphenicol. The BS sample did not show any antimicrobial activity, while chloramphenicol showed inhibition at 70 µg·mL^−1^.

The MIC of OGF demonstrated inhibition against five bacteria, showing strong inhibition against both Gram-negative and Gram-positive bacteria, such as *S. choleraesuis*, *E. coli*, and *S. aureus* (1000 µg·mL^−1^), and moderate inhibition against *L. innocua* and *B. cereus* (2000 µg·mL^−1^). However, OGF did not inhibit *P. aeruginosa* (>2000 µg·mL^−1^). Additionally, the MBC of OGF exhibited a bactericidal effect, inhibiting 99.9% of bacterial growth for *E. coli*, *B. cereus*, and *S. aureus* at 2000 µg·mL^−1^. OGF demonstrated that sensitivity did not significantly vary between Gram-positive and Gram-negative bacteria. Essential oils are generally more effective against Gram-positive bacteria due to the absence of a protective outer membrane with a double lipid layer, which is present in Gram-negative bacteria. However, this efficacy can vary, with EO showing different levels of activity against both Gram-positive and Gram-negative bacteria [2].

These findings corroborate those of Melo et al. [49], who studied the antimicrobial activity of *O. gratissimum* EO against isolates of *S. aureus* (4) and *E. coli* (4). The MIC was 1000 µg·mL^−1^ against all isolates, with MBC values ranging from 1000 to 2000 µg·mL^−1^. Similarly, the results by Chimnoi et al. [16] showed that OG essential oil inhibited *S. aureus*, *E. coli*, *Salmonella* Typhimurium, and *Shigella flexneri*, with MIC values ranging from 1000 to 2000 µg·mL^−1^. Likewise, Sartoratto et al. [50] analyzed OG essential oil against nine bacteria, finding MIC values of 300 µg·mL^−1^ for *E. faecium*, 600 µg·mL^−1^ for *Salmonella choleraesuis*, and 1000 µg·mL^−1^ for *S. aureus* and *B. subtilis*. However, *E. coli*, *R. equi*, and *M. luteus* showed MIC values above 2000 µg·mL^−1^, and no activity was observed against *P. aeruginosa*, *E. faecium*, and *S. epidermidis*.

The antimicrobial activity of *Ocimum gratissimum* essential oil is primarily attributed to its major component, eugenol [1]. Eugenol, which is also present in 85% of clove essential oil, can disrupt bacterial cell walls through its hydroxyl group, which binds to proteins and inhibits enzymatic actions [2]. Additionally, eugenol can denature proteins, rupture the cytoplasmic membrane, and alter cell permeability, leading to cellular damage and bacterial death [51]. Plants with high eugenol content exhibit strong antibacterial potential against both Gram-positive and Gram-negative bacteria [16]. However, the amount of eugenol and its antimicrobial effect can vary depending on climatic conditions and seasons [10].

The results for OGE microparticles demonstrated greater inhibition at lower concentrations compared to OGF, showing strong inhibition at 500 µg·mL^−1^ against *S. choleraesuis*, *E. coli*, and *S. aureus*, and moderate inhibition at 1000 µg·mL^−1^ against *L. innocua* and *B. cereus*. Additionally, OGE showed inhibition at 2000 µg·mL^−1^ against the Gram-negative bacterium *P. aeruginosa*. The MBC results were also significant, with minimum bactericidal effect values observed for OGE microparticles against all six bacteria studied, ranging from 1000 to 2000 µg·mL^−1^. These findings confirm that OGE microparticles performed better than OGF due to the lower MIC and MBC values observed in this study.

Few studies have reported on the encapsulation of OG essential oil and its effect on antimicrobial activity; however, similar work has been considered. In the study by Araujo et al. [1], nanoemulsions of OG EO were prepared using cashew gum, and their efficacy was evaluated against three bacteria. MIC values were 1300 µg·mL^−1^ for non-encapsulated EO and 650 µg·mL^−1^ for the nanoemulsion. It was observed that the MBC of the nanoemulsions was lower for Gram-negative bacteria (1300 µg·mL^−1^) compared to Gram-positive bacteria (5180 µg·mL^−1^). Thus, the nanoemulsions demonstrated superior antibacterial activity against *S. aureus*, *E. coli*, and *Salmonella enterica* compared to non-encapsulated EO.

Donsì and Ferrari [61] highlight that encapsulation in nanoemulsions enhances the dispersibility and physicochemical stability of essential oils and intensifies their interaction with microbial cells. This process facilitates the approach of essential oil droplets to the cell membrane, boosting antimicrobial efficacy through membrane disruption and interference with cellular integrity. Similarly, Yoplac [62] observed that the microencapsulation of citral with dextrin via spray-drying increases the antimicrobial efficacy of citral through three mechanisms: increasing the contact surface area between the microparticle and bacteria, gradual release of citral, and electrostatic interaction with bacterial cell walls, concentrating citral at the site of action and enhancing its efficiency.

## 4. Conclusions

In this study, the encapsulation process of *Ocimum gratissimum* essential oil did not alter its main constituents or its antioxidant and antimicrobial activities. The essential oil microparticles of OG were able to inhibit the pathogenic bacteria, including *S. aureus*, *B. cereus*, *L. innocua*, *E. coli*, *S. Choleraesuis*, and *P. aeruginosa*, in addition to exhibiting high antioxidant capacity. The main component of OG essential oil identified by GC–MS was eugenol, a compound widely recognized in the literature for its potent antimicrobial and antioxidant activities, which was confirmed in this study. A novel composition of biopolymers, maltodextrin, inulin, and cashew gum was used in equal proportions in the encapsulation process via the spray-drying method. The selection of these polymers may offer a functional and cost-effective alternative for encapsulating essential oils, highlighting the use of cashew gum and inulin. Future research is necessary to analyze different concentrations of these biopolymers, study the emulsion properties, and evaluate their application in food matrices to extend product shelf life. Therefore, OGE emerges as a natural alternative with antioxidant and antimicrobial properties, showing potential for use in the food, pharmaceutical, and cosmetic industries, as well as a flavoring agent.

## Figures and Tables

**Figure 1 foods-13-03122-f001:**
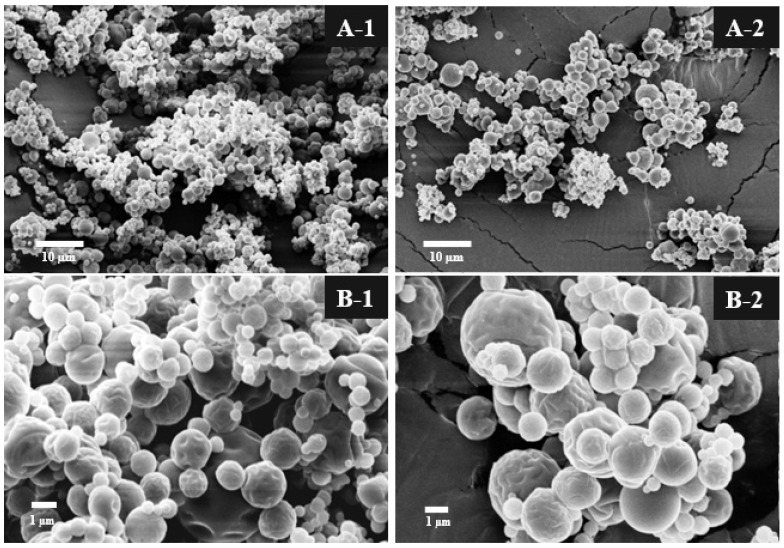
SEM images of OGE microparticles: (**A-1**,**A-2**) (2500× magnification, 10 μm scale bar), (**B-1**,**B-2**) (10,000× magnification, 1 μm scale bar).

**Figure 2 foods-13-03122-f002:**
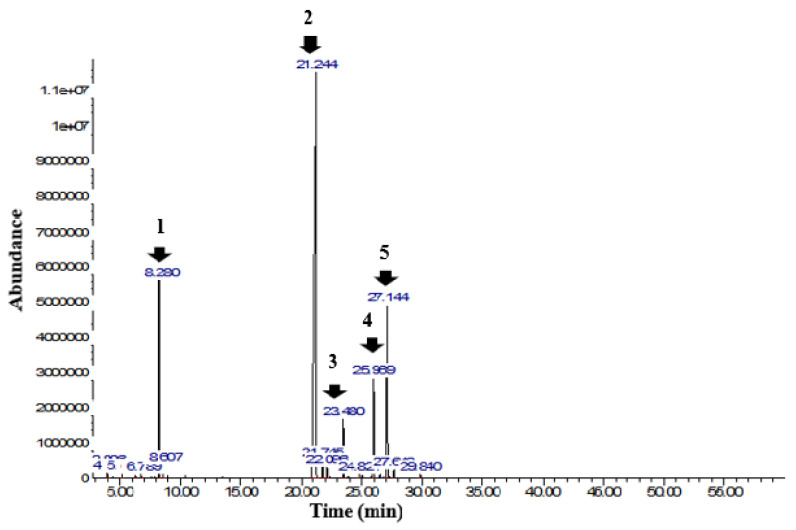
GC–MS chromatogram of *Ocimum gratissimum* essential oil. Peak 1: cis-β-Ocimene; Peak 2: Eugenol; Peak 3: β-Caryophyllene; Peak 4: Germacrene-D; Peak 5: Farnesene.

**Figure 3 foods-13-03122-f003:**
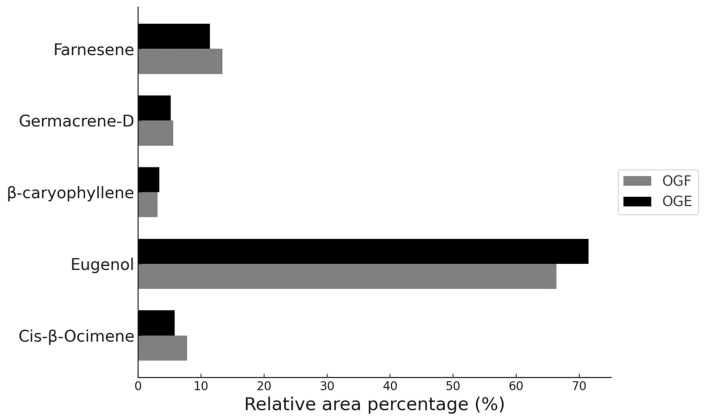
Variations in the relative area (%) of major chemical compounds in *Ocimum gratissimum* essential oil.

**Figure 4 foods-13-03122-f004:**
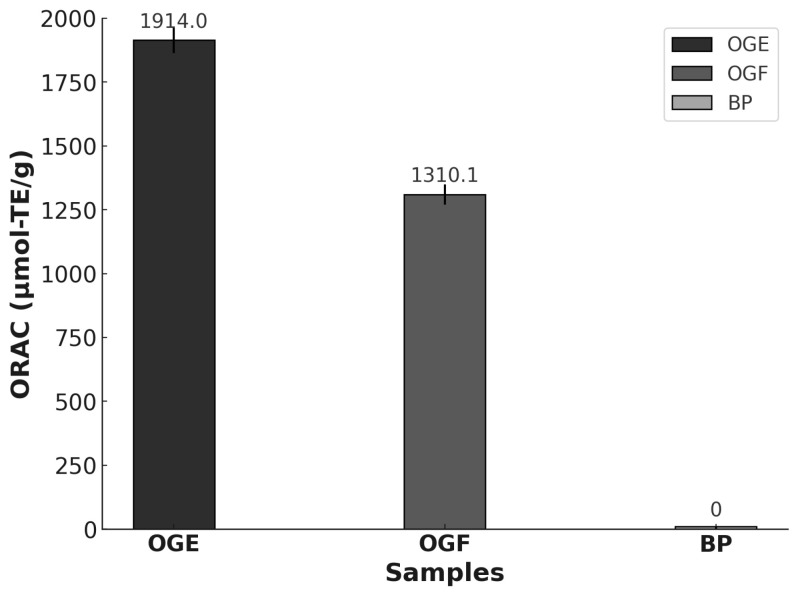
Antioxidant activity of OGF and OGE.

**Figure 5 foods-13-03122-f005:**
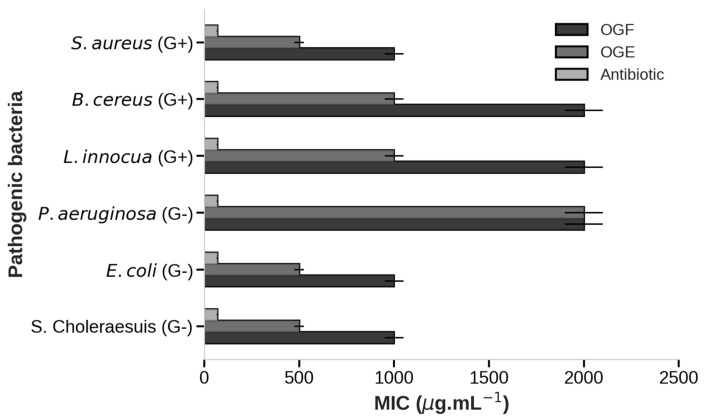
Growth inhibition of six pathogenic bacteria by OGF, OGE, and an antibiotic.

**Table 1 foods-13-03122-t001:** Characteristics of *Ocimum gratissimum* essential oil microparticles.

Properties	OGE *
Moisture g/100 g (dry basis)	3.91% ± 0.11
EE (%)	45.2% ± 0.4
aw	0.25 ± 0.02
D_4.3_ (µm)	7.32 ± 0.81
Span	1.25 ± 0.05
D_10_ (µm)	3.27 ± 0.04
D_50_ (µm)	6.29 ± 0.17
D_90_ (µm)	11.11 ± 0.48

* Values represent the mean of triplicates ± standard deviation.

**Table 2 foods-13-03122-t002:** Identification of compounds in OGF and OGE microparticles.

Compound Name		OGF			OGE		
		RT	RI	%A	RT	RI	%A
1	Ethylbenzene	3.88	858	0.3	-	-	-
2	Para- xylene	4.02	867	0.1	-	-	-
3	α—thujene	5.16	925	0.1	-	-	-
4	Sabinene	6.33	971	0.1	-	-	-
5	Myrcene	6.79	989	0.1	-	-	-
6	Cis-β-Ocimene	8.28	1035	7.8	8.24	1034	5.8
7	Trans-Ocimene	8.61	1045	0.5	8.56	1043	0.4
8	Eugenol	21.25	1362	66.4	21.31	1363	71.5
9	Copaene	21.74	1374	0.9	21.71	1373	1
10	Bourbonene	22.09	1382	0.6	22.06	1381	0.6
11	β-caryophyllene	23.48	1416	3.1	23.45	1415	3.4
12	Humulene	24.82	1450	0.2	-	-	-
13	Germacrene-D	25.97	1478	5.6	25.93	1477	5.2
14	Farnesene	27.15	1508	13.4	27.08	1506	11.4
15	δ—Cadinene	27.65	1521	0.5	27.6	1519	0.7
16	Caryophyllene oxide	29.84	1578	0.3	-	-	-

RI = retention index; RT = retention time (min); %A = relative area percentage; (-) = compounds absent or below detection limits.

**Table 3 foods-13-03122-t003:** Minimum Bactericidal Concentration values of OGF and OGE in µg·mL^−1^ against pathogenic bacteria.

Pathogenic Bacteria	OGF	OGE
*Salmonella choleraesuis* (G−)	>2000	1000
*Escherichia coli* (G−)	2000	1000
*Pseudomona aeruginosa* (G−)	>2000	2000
*Listeria innocua* (G+)	>2000	2000
*Bacillus cereus* (G+)	>2000	2000
*Staphylococcus aureus* (G+)	2000	1000

Gram-positive (G+); Gram-negative (G−).

## Data Availability

The original contributions presented in this study are included in this article. Further inquiries can be directed to the corresponding authors.

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
