# Peer review of "Impact of Microencapsulation on Ocimum gratissimum L. Essential Oil: Antimicrobial, Antioxidant Activities, and Chemical Composition"

_foods, 2024, doi:10.3390/foods13193122_

Round 1

Reviewer 1 Report

Comments and Suggestions for Authors

The manuscript primarily investigates the encapsulation of Ocimum gratissimum L. essential oil with maltodextrin, cashew gum, and inulin as biopolymers, evaluating the major components and their antioxidant and antimicrobial properties. The overall research aim is clear, the experimental design is sound, the logic is coherent, and the content is complete. However, there are certain issues, especially in writing, that need further improvement by the authors.

 1. The title is not concise enough. It is overly lengthy. Notably, the first part of the title mentions "antimicrobial," while the latter part refers to "its effect against foodborne pathogenic bacteria." In fact, the “antimicrobial” effect studied is precisely against the six foodborne pathogenic bacteria.

 2. Microencapsulation is a central aspect of this study. While I do not deny that this technique may not have been previously applied to this specific essential oil, research on microencapsulation of essential oils certainly exists. The use of microencapsulation in this study was inspired by related literature, so references about essential oil microencapsulation should be added to the second paragraph of "1. Introduction."

 3. In the third paragraph of "1. Introduction," the three biopolymers are introduced and immediately followed by the study's objectives. This logic is flawed. The study does not solely focus on investigating these three biopolymers; it also addresses topics from the first and second paragraphs. Therefore, the study's content and objectives should be presented in a separate paragraph.

 4. In "3. Results and Discussion," sections 3.2 and 3.3 deal with "Ocimum gratissimum essential oil" and "OGE" respectively, both focusing on "Chemical characterization." Thus, separating them is inappropriate, and grouping them would aid readers in understanding variations between them more clearly. Similarly, this applies to sections 3.4 and 3.5. It is preferable to present the control and OGE experimental results together before progressing to "Discussion," enabling readers to easily compare them, enhancing cohesion. Therefore, these sections require rewriting by the authors.

 5. A common writing issue in the manuscript is integrating "citation markers" directly into sentences, such as “This moisture level was similar to those reported by [36],” “The batch of cashew gum was purified according to the methodology described by [28],” “[13] identified 37 compounds, with five main ones……” This occurs frequently. Instead, the author's name should replace this format. Please thoroughly review and correct similar issues throughout the manuscript.

 6. Many figures are too rudimentary; most are not standardized. Figures 1 and 3 should not directly use screenshots or copies from software for the manuscript, as they lack clarity, and their font sizes and colors are inappropriate. For Figure 2, only the scale should be provided, placed directly on the figure, and remove any unrelated information, as it too lacks clarity. Figures 4, 5, and 6 should have their backgrounds removed. Fonts, font sizes, and colors at the same level in all figures should be consistent; also, the bars in Figure 5 are too thick. In summary, the current figures are of poor quality. Additionally, any unrelated information in the figures should be removed. The placement and substance of figure titles do not conform to standard norms, as they are not genuine titles; hence, the figure titles require substantial revision.

 7. What do the differently colored lines in Figure 1 represent?

 8. For words with abbreviations, provide the full term only upon first occurrence, and use the abbreviation thereafter.

 9. Line 265: What does the "1:1 ratio" refer to?

Comments on the Quality of English Language

It's acceptable, there are no major issues

Author Response

For research article

Response to Reviewer X Comments

Thank you for your feedback and for highlighting the strengths of our manuscript. We acknowledge that there are areas, particularly in our writing, that require further improvement. We have thoroughly reviewed the manuscript to address these issues and have implemented the necessary revisions to enhance its quality.

  1. Questions for General Evaluation

Reviewer’s Evaluation

Does the introduction provide sufficient background and include all relevant references? 

(X) Must be improved

Response and Revisions

Additional references have been considered to enhance the introduction, and all the suggestions mentioned in point 3 (Point-by-Point Response to Comments and Suggestions for Authors) have been addressed.

  1. Point-by-point response to Comments and Suggestions for Authors

Comments 1: “The title is not concise enough. It is overly lengthy. Notably, the first part of the title mentions "antimicrobial," while the latter part refers to "its effect against foodborne pathogenic bacteria." In fact, the “antimicrobial” effect studied is precisely against the six foodborne pathogenic bacteria”.

Response 1: Thank you for pointing this out. We agree with your comment and have consequently revised the title of the manuscript from “Evaluation of the chemical composition, antimicrobial and antioxidant activities of free and microencapsulated Ocimum gratissimum L. essential oil, and its effect against foodborne pathogenic bacteria” to 'Impact of microencapsulation on the chemical composition and antimicrobial and antioxidant activities of Ocimum gratissimum L. Essential Oil', as indicated in lines 2–5.

Comments 2: “Microencapsulation is a central aspect of this study. While I do not deny that this technique may not have been previously applied to this specific essential oil, research on microencapsulation of essential oils certainly exists. The use of microencapsulation in this study was inspired by related literature, so references about essential oil microencapsulation should be added to the second paragraph of "1. Introduction."

Response 2: Thank you for your observation. We agree that there are previous studies on the microencapsulation of essential oils, and including these references can provide clearer context. Therefore, we have added relevant references to the second paragraph of the "Introduction," which can be found in lines 61-64.

Comments 3: “In the third paragraph of "1. Introduction," the three biopolymers are introduced and immediately followed by the study's objectives. This logic is flawed. The study does not solely focus on investigating these three biopolymers; it also addresses topics from the first and second paragraphs. Therefore, the study's content and objectives should be presented in a separate paragraph”.

Response 3:  We agree with the comment. Therefore, we have allocated the study objective to a separate paragraph on lines 85-88.

Comments 4:  In "3. Results and Discussion," sections 3.2 and 3.3 deal with "Ocimum gratissimum essential oil" and "OGE" respectively, both focusing on "Chemical characterization." Thus, separating them is inappropriate, and grouping them would aid readers in understanding variations between them more clearly. Similarly, this applies to sections 3.4 and 3.5. It is preferable to present the control and OGE experimental results together before progressing to "Discussion," enabling readers to easily compare them, enhancing cohesion. Therefore, these sections require rewriting by the authors.

Response 4: We greatly appreciate your observation. We would like to clarify that, initially, sections 3.2 and 3.3, which address chemical characterization, were presented together. However, we decided to separate them to facilitate readers’ understanding. Section 3.2 was dedicated to identifying the chemical composition of the essential oil and highlighting the importance of its major compound, eugenol, with the intention of emphasizing the relevance of this plant and informing readers about the potential of Ocimum gratissimum essential oil.

In section 3.3, our goal was to clearly explain the reasons behind the loss of compounds during the microencapsulation process. We believe it is important to address this issue in detail, as few authors delve into this specific aspect. Therefore, the separation of these sections was aimed at providing greater clarity and understanding of the different points discussed.

Additionally, upon revisiting section 3.4 on antioxidant activity, we observed that the results are presented prior to the discussion, as suggested. In section 3.5, which addresses antibacterial activity, the results are presented in this manner because we considered the MIC and MBC values for both the free and encapsulated essential oils. In order to facilitate comprehension, we first sought to highlight the antimicrobial action of the essential oil and then focus on the results of the microparticles, followed by their respective discussions.

However, if it is still necessary to implement the suggested changes, please let us know, and we will be happy to make the adjustments.

Comments 5:  “A common writing issue in the manuscript is integrating "citation markers" directly into sentences, such as “This moisture level was similar to those reported by [36],” “The batch of cashew gum was purified according to the methodology described by [28],” “[13] identified 37 compounds, with five main ones……” This occurs frequently. Instead, the author's name should replace this format. Please thoroughly review and correct similar issues throughout the manuscript”.

Response 5: Thank you for the observation. I followed the reviewer's recommendations and carefully revised the manuscript, replacing the citation markers with the authors' surnames as suggested. However, I kept some citations in numerical format, particularly in the introduction, as it is a requirement of the journal, which does not request the inclusion of the author's name in this section. I hope these revisions meet the expectations of the editorial team.

Comments 6:   “Many figures are too rudimentary; most are not standardized. Figures 1 and 3 should not directly use screenshots or copies from software for the manuscript, as they lack clarity, and their font sizes and colors are inappropriate. For Figure 2, only the scale should be provided, placed directly on the figure, and remove any unrelated information, as it too lacks clarity. Figures 4, 5, and 6 should have their backgrounds removed. Fonts, font sizes, and colors at the same level in all figures should be consistent; also, the bars in Figure 5 are too thick. In summary, the current figures are of poor quality. Additionally, any unrelated information in the figures should be removed. The placement and substance of figure titles do not conform to standard norms, as they are not genuine titles; hence, the figure titles require substantial revision”.

Response 6: We greatly appreciate the detailed feedback regarding the figures. We have already made the necessary corrections to ensure that the figures are standardized and meet the clarity requirements. Specifically:

  • Figure 1 was removed, and the numerical data were included directly to facilitate the understanding of the results. Consequently, all figure numbers and identifications in the manuscript were modified to reflect this change.
  • The image quality of Figure 3 was improved. We had some concerns regarding the use of the image provided by the software, as it has been accepted in previous publications, and we hope it can be considered in this context.
  • In Figure 2, the scale was adjusted directly within the figure, and any unrelated information was removed to enhance its clarity.
  • We removed the backgrounds from Figures 4, 5, and 6 and ensured that all fonts, font sizes, and colors are consistent across the figures. Additionally, we thinned the bars in Figure 5 for better presentation.

Furthermore, we revised the figure titles to ensure they comply with standard norms and accurately reflect the content of each figure.

We thank you again for the feedback and have already implemented the recommended improvements to ensure the quality of the figures. We hope that the modifications meet the editorial team's expectations.

Comments 7:  “What do the differently colored lines in Figure 1 represent?”

Response 7: The different colored lines in Figure 1 represent the various groups of samples analyzed in the experiment. Each color corresponds to a specific group, allowing for a clearer comparison of the results obtained for each. However, after careful consideration, we have decided to omit Figure 1, which illustrated the differential distribution of particle size (µm), and instead rely on the numerical data for the analysis. This adjustment was made to enhance clarity and ensure the results are more easily understood by the reader.

Comments 8:  “For words with abbreviations, provide the full term only upon first occurrence, and use the abbreviation thereafter”.

Response 8: We appreciate your observation. We have reviewed the manuscript to ensure that all terms with abbreviations are properly presented, as requested.

Comments 9:   Line 265: What does the "1:1 ratio" refer to?

Response 9:

Line 265 was reviewed, and indeed there was a missing mention of one of the polymers used. The 1:1 ratio refers to the presence of a combination of two polymers, GC and IN. In this line, the biopolymer inulin (IN) was added.

  1. Additional clarifications

Reviewer 2 Report

Comments and Suggestions for Authors

The authors selected maltodextrin, cashew gum, and inulin as microcapsule materials to prepare microcapsules containing Ocimum gratissimum essential oil by spray-drying method, meanwhile investigated their antimicrobial and antioxidant effects.The analysis identified eugenol as the main compound both before and after micro encapsulation. The OGE microparticles demonstrated high inhibitory and bactericidal effects against S. aureus, S. Choleraesuis and E. coli. Most of the scientific work had been explained with concrete words and figures. Therefore, I recommended some corrections need to be made listed as follows:

1.For the title, it seemed a little long and was suggested to draw up a new one according to the content.

2. For part 2.2.3., the author mentioned ‘Maltodextrin, cashew gum, and inulin were weighed in equal parts (1:1:1)’,why was this ratio chosen here?

3. For Figure 1, the figure showed curves of different colors, please explain in the note.

4. The author mentioned ‘observed that drying temperature (120, 155, and 190 °C) influenced the formation of smooth or rough surfaces’, please present the results of this research.

5. For Figure 5&6, how many times did the author repeat the experiment? No error lines were shown in the figure.

6. For the conclusion part, it is suggested to rewrite, which should have the review of this paper and the prospect of the future.

7. For references, it is recommended to include some of the latest literature.

Author Response

For research article

Response to Reviewer X Comments

Thank you for your comprehensive review of our manuscript and for highlighting its strengths. We have carefully addressed the recommended corrections and made the necessary revisions to further enhance the quality and clarity of our work.

  1. Questions for General Evaluation

Reviewer’s Evaluation

Does the introduction provide sufficient background and include all relevant references? (X) Can be improved   

Is the research design appropriate? (X) Can be improved   

Are the methods adequately described? (X) Can be improved   

Are the results clearly presented? (X) Can be improved   

Are the conclusions supported by the results? (X) Can be improved   

Response and Revisions

Changes were made to the introduction, methods, and figures, and all the suggestions in point 3 have also been addressed.

  1. Point-by-point response to Comments and Suggestions for Authors

Comments 1: “ For the title, it seemed a little long and was suggested to draw up a new one according to the content.”.

Response 1: Thank you for pointing this out. We agree with your comment and have consequently revised the title of the manuscript from “Evaluation of the chemical composition, antimicrobial and antioxidant activities of free and microencapsulated Ocimum gratissimum L. essential oil, and its effect against foodborne pathogenic bacteria” to 'Impact of microencapsulation on the chemical composition and antimicrobial and antioxidant activities of Ocimum gratissimum L. Essential Oil', as indicated in lines 2–4.

Comments 2: “For part 2.2.3., the author mentioned ‘Maltodextrin, cashew gum, and inulin were weighed in equal parts (1:1:1)’,why was this ratio chosen here?"

Response 2: Thank you for your question. The 1:1:1 ratio of maltodextrin, cashew gum, and inulin was chosen to ensure a balanced distribution of the polymers in the emulsion formulation. Additionally, this proportion has been used in previous studies, such as Fernandes et al. (2016), cited on line 265, which demonstrated improved encapsulation efficiency, making it an appropriate starting point for our study. Furthermore, our research group specifically studies cashew gum, and this study focused on its combination with other biopolymers.

Comments 3: “For Figure 1, the figure showed curves of different colors, please explain in the note”.

Response 3:  The different colored lines in Figure 1 represent the various groups of samples analyzed in the experiment. Each color corresponds to a specific group, allowing for a clearer comparison of the results obtained for each. However, after careful consideration, we have decided to omit Figure 1, which illustrated the differential distribution of particle size (µm), and instead rely on the numerical data for the analysis. This adjustment was made to enhance clarity and ensure the results are more easily understood by the reader.

Comments 4:  The author mentioned ‘observed that drying temperature (120, 155, and 190 °C) influenced the formation of smooth or rough surfaces’, please present the results of this research.

Response 4: We appreciate this observation. We have noticed that the information related to the drying temperatures (120, 155, and 190 °C) may not be clearly connected to the explanation provided by the cited authors. We have made the necessary modifications between lines 309 and 315 to enhance the understanding of the results and discussions presented in the study by Beirão-da-Costa et al. (2013).

Comments 5:  “For Figure 5&6, how many times did the author repeat the experiment? No error lines were shown in the figure”.

Response 5: The experiment related to Figures 5 and 6 was repeated three times to ensure the reproducibility of the results. However, we acknowledge that error bars were not included in the figures. We have already corrected this oversight and added the appropriate error bars to reflect the variability in the data. Additionally, we made several modifications to the manuscript, and as Figure 1 was removed, the numbering of the figures has been adjusted accordingly. We appreciate your observation and hope that these changes meet the expectations.

Comments 6:   “For the conclusion part, it is suggested to rewrite, which should have the review of this paper and the prospect of the future”.

Response 6: We appreciate the suggestion regarding the conclusion. We have rewritten it to include a review of the key points discussed in the article, as well as a perspective on future research and applications related to the topic. The necessary changes have been made to ensure that the conclusion reflects both the findings and the potential for future advancements in this field.

Comments 7:  “For references, it is recommended to include some of the latest literature”.

Response 7: We appreciate the recommendation regarding the references. We have already reviewed the bibliography and included more recent literature, particularly in the introduction, to ensure that the manuscript is up to date with the latest research in the field. We hope these updates meet the expectations of the editorial team.

  1. Additional clarifications

Reviewer 3 Report

Comments and Suggestions for Authors

Along the manuscript: Every time you use the reference as “by [Ref]”, please, indicate before the author surname, as “by author surname/author surname et al. [Ref]”.

Line 99: indicate (v/v) (m/v) (m/m)?

Line 100: How was it filtered? Was filtered by gravity? Under vacuum?. Indicate how was filtered.

Line 101: What does GL means? Indicate

Line 102: How was it filtered? Indicate.

Ine 108-109: Why was not used dried leaves for the extraction method?. Indicate the ratio of mass/solvent used for the extraction procedure.

Line 301: Why was not used higher amplifications for SEM analyses? I agree that no cracks look like to be in the particles showing an apparent continuous surface, however, clefts can be observed. Higher augmentations will provide better identification of particle surface.

Author Response

For research article

Response to Reviewer X Comments

Thank you for your comprehensive review of our manuscript. We have thoroughly addressed the suggested corrections and implemented the necessary revisions to further enhance the quality and clarity of our work.

  1. Questions for General Evaluation

Reviewer’s Evaluation

Are the methods adequately described? (X) Can be improved   

  • Along the manuscript: Every time you use the reference as “by [Ref]”, please, indicate before the author surname, as “by author surname/author surname et al. [Ref]”.

Response and Revisions

Thank you for the observation. I followed the reviewer's recommendations and carefully revised the manuscript, replacing the citation markers with the authors' surnames as suggested. However, I kept some citations in numerical format, particularly in the introduction, as it is a requirement of the journal, which does not request the inclusion of the author's name in this section. I hope these revisions meet the expectations of the editorial team.

  1. Point-by-point response to Comments and Suggestions for Authors

Comments 1: “Line 99: indicate (v/v) (m/v) (m/m)?”

Response 1: We appreciate this observation. As suggested, we have specified the concentration of the solution (w/v) ratio on line 109, indicating the exudate: water ratio.

Comments 2: “Line 100: How was it filtered? Was filtered by gravity? Under vacuum?. Indicate how was filtered”.

Response 2: We have added the description of the gravity filtration technique on line 109.

Comments 3: “Line 101: What does GL means? Indicate”.

Response 3:  The meaning of "GL" was indicated in lines 111 and 112. “ Subsequently, this solution was mixed with 96 % ethanol (96° Gay-Lussac) at a 1:8 (v/v) ratio (solution: ethanol)”

Comments 4:  Line 102: How was it filtered? Indicate.

Response 4: We appreciate this observation. As indicated on line 112, the sample was filtered again using the gravity filtration technique: "to promote gum precipitation, which was immediately filtered again using gravity filtration."

Comments 5:  Iine 108-109: Why was not used dried leaves for the extraction method?. Indicate the ratio of mass/solvent used for the extraction procedure.

Response 5: Information regarding the mass/solvent ratio used during the extraction process has been added on lines 119 and 120: “The extraction was carried out from fresh leaves by hydrodistillation at a 1:3 (w/v) ratio of plant material to distilled water, using a Clevenger-type apparatus for 3 h”.

Regarding the question of why dried leaves were not used for the essential oil extraction, fresh samples were chosen to maintain consistency with the procedures used in previous studies conducted by the team of researchers and professors at the CPQBA-UNICAMP research center, as outlined in the methodology cited in section 2.2.2.

Comments 6:   “Line 301: Why was not used higher amplifications for SEM analyses? I agree that no cracks look like to be in the particles showing an apparent continuous surface, however, clefts can be observed. Higher augmentations will provide better identification of particle surface”.

Response 6: We appreciate your observation. We chose not to use higher magnifications in the scanning electron microscopy (SEM) analyses because our primary objective was to evaluate the general morphology and surface integrity of the microparticles. The magnifications used were considered sufficient to demonstrate the absence of visible cracks and to show a continuous surface. However, we acknowledge that higher magnifications could provide a more detailed identification of potential surface fissures, and we will consider this suggestion in future studies.

  1. Additional clarifications

I would like to kindly inform you that the title of the manuscript has been changed, following the suggestions of two of the reviewers. I hope this adjustment aligns with the expectations of the editorial team.

Reviewer 4 Report

Comments and Suggestions for Authors

The research article entitled „Evaluation of the chemical composition, antimicrobial and antioxidant activities of free and microencapsulated Ocimum gratissimumL. essential oil, and its effect against foodborne pathogenic bacteria“ provides valuable insights into the chemical profile of Ocimum gratissimum L. essential oil in free and encapsulated form, as well as their antioxidant and antibacterial activity. The paper is generally well-written, with a formal and scientific tone appropriate for an academic audience. However, the article needs revisions to improve clarity and depth of analysis.

1. Missing * for Angela P. Flores Granados1, as this Author is one of the corresponding authors.

2. The keywords section should include the following words: Ocimum gratissimum L.;  essential oil; microencapsulation.

3. In the Introduction, lines 38-48 should address a more detailed explanation of the natural food additives and microencapsulation. Besides essential oil, what else can be used as natural additives? What are the benefits of microencapsulation?

4. In the Introduction, the aim of the research article (lines 77-79) differs from the aim stated in the abstract section (lines 20-22).

5. What was the model of used rotary evaporator?

6. The role of each polymer used for emulsion production is well described. However, did you do the preliminary study regarding the maltodextrin, cashew gum, and inulin ratio in the formulation of the emulsion?

7. Abbreviation of maltodextrin (MA), cashew gum (CG), and inulin (IN) should be defined when they appear for the first time (line 24; lines 65, 68 and 73 instead of line 230). There are similar errors throughout the text, please correct this.

Instructions for Authors: Acronyms/Abbreviations/Initialisms should be defined the first time they appear in each of three sections: the abstract; the main text; the first figure or table. When defined for the first time, the acronym/abbreviation/initialism should be added in parentheses after the written-out form.

8. Table 1 is not mentioned anywhere in the text.

9. In Table 2. the wrong abbreviation was used in columns for the retention index and time.

10. In table 3. there are words in Portuguese ( livre, encapsulado).

11. Why is this study limited to foodborne pathogenic bacteria, when Ocimum gratissimum L.; essential oil posses antifungal activity (https://doi.org/10.1016/j.foodchem.2022.134087, https://doi.org/10.1016/j.prenap.2024.100065).

Author Response

For research article

Response to Reviewer X Comments

We are grateful for your positive feedback on our manuscript. We acknowledge your suggestion to enhance the clarity and depth of our analysis, and we will thoroughly review the manuscript to implement the necessary changes that improve its overall comprehension and quality.

  1. Questions for General Evaluation

Reviewer’s Evaluation

Does the introduction provide sufficient background and include all relevant references? (X) Can be improved   

Is the research design appropriate? (X) Can be improved   

Response and Revisions

Additional references have been considered to enhance the introduction, and all the suggestions mentioned in point 3 (Point-by-Point Response to Comments and Suggestions for Authors) have been addressed.

  1. Point-by-point response to Comments and Suggestions for Authors

Comments 1: “ Missing * for Angela P. Flores Granados1, as this Author is one of the corresponding authors”.

Response 1: Thank you for bringing this matter to our attention. We have revised the manuscript to include the missing asterisk (*) next to the name of Angela P. Flores Granados, indicating her role as one of the corresponding authors.

Comments 2: “The keywords section should include the following words: Ocimum gratissimum L.;  essential oil; microencapsulation."

Response 2: Thank you for your suggestion regarding the keywords. We would like to point out that the terms "Ocimum gratissimum L.," "essential oil," and "microencapsulation" are already present in the title of the manuscript, as we aimed to clearly convey the main focus of our study. However, if you believe additional keywords are necessary to further enhance the manuscript's discoverability, we are open to considering them.

Comments 3: “In the Introduction, lines 38-48 should address a more detailed explanation of the natural food additives and microencapsulation. Besides essential oil, what else can be used as natural additives? What are the benefits of microencapsulation?”.

Response 3:  We appreciate your observation. We have revised the introduction to provide a more detailed explanation of natural food additives in lines 44-46, and the process of microencapsulation of essential oils in lines 61-65. Regarding the benefits of microencapsulation, we have highlighted this technique in lines 59-61, emphasizing how it protects active compounds, increases their stability, and improves the controlled release of natural additives.

Comments 4:  In the Introduction, the aim of the research article (lines 77-79) differs from the aim stated in the abstract section (lines 20-22).

Response 4: Thank you for pointing out that the objectives in the Introduction and the Abstract differ. We have reviewed and revised these sections to ensure they are aligned and coherently describe the purpose of the study. We appreciate your observation in helping to improve the clarity of the manuscript.

Comments 5:  “What was the model of used rotary evaporator?”.

Response 5: The rotary evaporator used in our study was the Rotavapor R-100, manufactured by Buchi, Switzerland. This information has been included in lines 148–149 of the manuscript.

Comments 6:   “The role of each polymer used for emulsion production is well described. However, did you do the preliminary study regarding the maltodextrin, cashew gum, and inulin ratio in the formulation of the emulsion?”.

Response 6: Thank you for your observation. We did not conduct a preliminary study specifically on the proportion of maltodextrin, cashew gum, and inulin in the emulsion formulation. The 1:1:1 ratio of these components was chosen to ensure a balanced distribution of the polymers in the emulsion. Moreover, this ratio has been used in previous studies, such as that of Fernandes et al. (2016), cited on line 269, which demonstrated greater encapsulation efficiency, and we considered it an appropriate starting point for our study. Additionally, our research group specifically works with cashew gum, and this study focused on its combination with other biopolymers. However, we believe that a more detailed analysis of the proportions of these polymers could be an interesting approach for future research.

Comments 7:  “Abbreviation of maltodextrin (MA), cashew gum (CG), and inulin (IN) should be defined when they appear for the first time (line 24; lines 65, 68 and 73 instead of line 230). There are similar errors throughout the text, please correct this”.

Instructions for Authors: Acronyms/Abbreviations/Initialisms should be defined the first time they appear in each of three sections: the abstract; the main text; the first figure or table. When defined for the first time, the acronym/abbreviation/initialism should be added in parentheses after the written-out form.

Response 7: Thank you for your comment. We have reviewed the manuscript and defined the abbreviations for maltodextrin (MD), cashew gum (CG), and inulin (IN) at their first appearance in each of the three sections: the abstract, the main text, and the first figure or table, in accordance with the instructions for authors. We have also corrected similar errors throughout the text to ensure consistency.

Comments 8: “Table 1 is not mentioned anywhere in the text”.

Response 8: Thank you for bringing this to our attention. We have reviewed the manuscript and added a reference to Table 1 on line 238 to ensure its proper mention and context.

Comments 9: “In Table 2. the wrong abbreviation was used in columns for the retention index and time”.

Response 9: Thank you for pointing out this error. We have reviewed Table 2 and corrected the abbreviations used for the retention index and retention time in the corresponding columns.

Comments 10: “In table 3. there are words in Portuguese ( livre, encapsulado)”.

Response 10: Thank you for pointing out this detail. We have reviewed Table 3 and replaced the Portuguese words ("livre" and "encapsulado") with their corresponding English terms ("free" and "encapsulated") to maintain language consistency throughout the manuscript.

Comments 11: Why is this study limited to foodborne pathogenic bacteria, when Ocimum gratissimum L.; essential oil posses antifungal activity (https://doi.org/10.1016/j.foodchem.2022.134087,https://doi.org/10.1016/j.prenap.2024.100065).

 Response 11: We appreciate your question. This study focuses on foodborne pathogenic bacteria, as our primary objective was to evaluate the effectiveness of Ocimum gratissimum L. essential oil against bacterial pathogens associated with foodborne illnesses. Additionally, another team of researchers from the same microbiology laboratory is studying the antifungal activity and other specific aspects of the essential oil. Nonetheless, we are aware of the documented antifungal activity of Ocimum gratissimum L. essential oil and its significance in this area.

  1. Additional clarifications

I would like to kindly inform you that the title of the manuscript has been changed, following the suggestions of two of the reviewers. I hope this adjustment aligns with the expectations of the editorial team.

Round 2

Reviewer 1 Report

Comments and Suggestions for Authors

The author has made significant revisions based on the review comments, but there are still some issues that need further modification:

  1. The title still needs further improvement. There are 3 "of" and 2 consecutive "and" in the title; moreover, "microencapsulation of" should be directly followed by "Ocimum gratissimum  L. essential oil"; additionally, the second "of" is clearly incorrect.

  2. The issue of "100 x" in Formula 1, which was mentioned in the first review, has not been addressed or modified by the author. My understanding is that this should be a multiplication symbol, but it is displayed as the English letter "x".

  3. Regarding the titles of all figures in the manuscript, the explanations for each subfigure are placed directly with the figure title. This format is likely inappropriate unless the journal has specific requirements.

  4. Concerning the request to "add literature related to essential oil microencapsulation in the second paragraph of the '1. Introduction' section," although the author has added a sentence, it is clearly too simple and needs further elaboration

Author Response

For research article

Response to Reviewer X Comments

The author has made significant revisions based on the review comments, but there are still some issues that need further modification:

  1. The title still needs further improvement. There are 3 "of" and 2 consecutive "and" in the title; moreover, "microencapsulation of" should be directly followed by "Ocimum gratissimum  L. essential oil"; additionally, the second "of" is clearly incorrect.

Response 1:  Thank you for your observations. I have made the necessary revisions to the title, addressing the excessive use of "of" and "and" and ensuring that "microencapsulation of" is now directly followed by "Ocimum gratissimum L. essential oil." I also corrected the second "of" as per your suggestion. I appreciate your feedback and hope the changes are now satisfactory.

  1. The issue of "100 x" in Formula 1, which was mentioned in the first review, has not been addressed or modified by the author. My understanding is that this should be a multiplication symbol, but it is displayed as the English letter "x".

Response 2:  Thank you for pointing out the issue in Formula 1 again. We apologize for not addressing it in the first revision. We have reviewed the document and made the necessary corrections to replace the "x" with the proper multiplication symbol. We greatly appreciate your attention to this detail and hope it is now correct.

  1. Regarding the titles of all figures in the manuscript, the explanations for each subfigure are placed directly with the figure title. This format is likely inappropriate unless the journal has specific requirements.

Response 3:  We understand that including the explanations for each subfigure directly in the figure titles may be inappropriate. We have revised the titles and separated the subfigure explanations to ensure they conform to the standard format.

  1. Concerning the request to "add literature related to essential oil microencapsulation in the second paragraph of the '1. Introduction' section," although the author has added a sentence, it is clearly too simple and needs further elaboration

Response 4:  We acknowledge that the sentence added regarding the microencapsulation of essential oils in the second paragraph of the Introduction was too simple. We have revised this section to expand and provide a more comprehensive context on the topic of microencapsulation. These adjustments can be seen in lines 57-71.

Reviewer 4 Report

Comments and Suggestions for Authors

Dear Authors,

Thank you for the revised manuscript!

Regarding the manuscript title,  the title „Impact of microencapsulation of the chemical composition and 2 antimicrobial and antioxidant activities of Ocimum gratissimum 3 L. essential oil“ fits better to the content of the manuscript.

Comments 2: “The keywords section should include the following words: Ocimum gratissimum L.;  essential oil; microencapsulation."

Response 2: Thank you for your suggestion regarding the keywords. We would like to point out that the terms "Ocimum gratissimum L.," "essential oil," and "microencapsulation" are already present in the title of the manuscript, as we aimed to clearly convey the main focus of our study. However, if you believe additional keywords are necessary to further enhance the manuscript's discoverability, we are open to considering them.

Comment 2: I will not insist in including these terms.

Comments 6:   “The role of each polymer used for emulsion production is well described. However, did you do the preliminary study regarding the maltodextrin, cashew gum, and inulin ratio in the formulation of the emulsion?”.

Response 6: Thank you for your observation. We did not conduct a preliminary study specifically on the proportion of maltodextrin, cashew gum, and inulin in the emulsion formulation. The 1:1:1 ratio of these components was chosen to ensure a balanced distribution of the polymers in the emulsion. Moreover, this ratio has been used in previous studies, such as that of Fernandes et al. (2016), cited on line 269, which demonstrated greater encapsulation efficiency, and we considered it an appropriate starting point for our study. Additionally, our research group specifically works with cashew gum, and this study focused on its combination with other biopolymers. However, we believe that a more detailed analysis of the proportions of these polymers could be an interesting approach for future research.

Comment 6: I am satisfied with the answer.

Comments 11: Why is this study limited to foodborne pathogenic bacteria, when Ocimum gratissimum L.; essential oil posses antifungal activity (https://doi.org/10.1016/j.foodchem.2022.134087,https://doi.org/10.1016/j.prenap.2024.100065).

Response 11: We appreciate your question. This study focuses on foodborne pathogenic bacteria, as our primary objective was to evaluate the effectiveness of Ocimum gratissimum L. essential oil against bacterial pathogens associated with foodborne illnesses. Additionally, another team of researchers from the same microbiology laboratory is studying the antifungal activity and other specific aspects of the essential oil. Nonetheless, we are aware of the documented antifungal activity of Ocimum gratissimum L. essential oil and its significance in this area.

Comment 11: In manuscript is highligted that they tested only on bacterial strains, therefore, they did all the planned investigation. I accept that manuscript stay in this form.

Regarding other responses to the comments (1, 3, 4, 5, 7, 9 and 10) , I am satisfied that they have been accepted. Also, after reading the revised manuscript, I do not have any other comments. I am pleased with the effort and improvements made to the manuscript.

Author Response

Thank you very much for your positive comments! We are pleased to know that you are satisfied with our responses and the improvements made to the manuscript. We appreciate your time and valuable observations, which have contributed to enhancing the quality of our work.